# CBCT-Based Retrospective Analysis of Posterior Superior Alveolar Artery Anatomy in a Saudi Population

**DOI:** 10.3390/diagnostics15232999

**Published:** 2025-11-26

**Authors:** Abdullah Alqhtani, Amirah Yahya Alhaidan, Asma Jumah Aloufi, Faten Sifran Alharbi, Lama Mohammed Alkahtani, Raghad Hatem Alamri, Mohamed Omar Elboraey

**Affiliations:** 1Department of Preventive Dental Sciences, College of Dentistry, Taibah University, Medinah 42353, Saudi Arabia; aqhtani@taibahu.edu.sa; 2College of Dentistry, Taibah University, Medinah 42353, Saudi Arabia; ameerhaidan@gmail.com (A.Y.A.); a.j.aloufi20@gmail.com (A.J.A.); dr.fatensafranalharbi@hotmail.com (F.S.A.); akahtani.lama2@gmail.com (L.M.A.); raghadalamri3737@gmail.com (R.H.A.); 3Department of Oral Medicine, Periodontology, Oral Diagnosis and Oral Radiology, Faculty of Dentistry, Tanta University, Tanta 31527, Egypt

**Keywords:** CBCT, PSAA, maxillary

## Abstract

**Background/Objectives**: Sinus elevation in the posterior maxilla carries a risk of hemorrhage due to injury of the posterior superior alveolar artery (PSAA). Accurate preoperative identification of the PSAA using cone-beam computed tomography (CBCT) can enhance surgical safety. This retrospective study evaluated the prevalence, location, diameter, and visibility of the PSAA in a Saudi population. **Methods**: A total of 117 CBCT scans (234 sinuses) obtained between 2022 and 2024 were analyzed. The PSAA’s visibility, diameter, and distances from the alveolar crest, sinus floor, medial wall, and sinus septa were measured. Associations with age, sex, smoking status, and the presence of sinus septa were statistically assessed. **Results**: The PSAA was identified in 98.3% of sinuses. Intraosseous and submucosal locations predominated in premolar and molar regions, respectively. Class A arteries (≤1 mm) were most frequent. Significant differences were found between premolar and molar regions in arterial location and distances from the alveolar crest and sinus walls (*p* < 0.001). Older individuals exhibited medial displacement of the artery in the molar region, and smokers showed significantly smaller diameters (*p* < 0.05). Sinus septa were associated with increased PSAA distances from the sinus floor and medial wall. **Conclusions**: The PSAA demonstrates high detectability and marked variability in position and caliber within this Saudi cohort. Recognition of these anatomical variations is essential for reducing complications during sinus-augmentation procedures.

## 1. Introduction

Placing dental implants in the posterior maxilla following tooth or teeth loss presents a significant clinical challenge. The reduced alveolar ridge height and maxillary sinus pneumatization, both of which compromise the available bone volume for implant placement, often result from progressive bone resorption and anatomical alterations after extraction [1,2].

The maxillary sinus is a pyramidal, air-filled cavity lined by the Schneiderian membrane and bounded superiorly by the orbital floor, medially by the lateral nasal wall, and inferiorly by the alveolar process [3]. Its close relationship to the posterior maxillary teeth and their supporting bone makes this region particularly susceptible to complications during surgical procedures. Anatomical variations such as sinus septa, mucosal thickening, and the course of the posterior superior alveolar artery (PSAA) can influence surgical access and the risk of intraoperative bleeding [4]. Consequently, sinus floor elevation techniques have become essential components in implant dentistry, particularly when the residual bone height is insufficient for achieving primary implant stability and long-term osseointegration [5]. Adequate bone quantity and quality are fundamental prerequisites for predictable implant success in this region.

One surgical technique that can be used is the lateral window technique, which was initially introduced by Tatum et al. (1986) [6] and is widely accepted for sinus augmentation. It involves raising the Schneiderian membrane after creating a bony window on the lateral wall of the maxillary sinus to allow for bone graft implantation [6]. This approach provides excellent visualization of the sinus cavity and allows for the placement of a large volume of graft material, making it suitable for cases with severely resorbed ridges [7].

Alternatively, a less invasive technique is the transcrestal sinus floor elevation described by Summers et al. (1994); this technique uses osteotomes through the alveolar crest to indirectly elevate the sinus membrane with minimal surgical trauma [8,9]. It is typically indicated when moderate bone height remains, reducing patient morbidity and postoperative discomfort. Advances in surgical instruments and the integration of piezoelectric devices have further refined this approach, improving precision and minimizing the risk of membrane perforation.

Despite the efficacy of both techniques, complications such as membrane perforation and hemorrhage may occur [4], especially when vital anatomical structures are not clearly identified [10]. Among these structures, the posterior superior alveolar artery (PSAA) is of critical significance, as its injury may complicate intraoperative management and impair graft healing.

The posterior superior alveolar artery (PSAA), an intraosseous vessel that anastomoses with the infraorbital artery and courses along the lateral wall of the maxillary sinus, is particularly vulnerable during bony window creation [11]. Injury to the PSAA during sinus membrane elevation can lead to significant bleeding and compromise graft placement and stability [4]. The artery’s anatomical variability, including its distance from the alveolar crest and sinus floor, emphasizes the need for individualized assessment before surgery. Preoperative cone-beam computed tomography (CBCT) is used to prevent these issues, and together with careful surgical planning, it is essential to identify anatomical variations and ensure successful outcomes. Accurate knowledge of the posterior superior alveolar artery’s (PSAA) position and course is therefore vital for enhancing the success rate of maxillary sinus bone grafts [12,13].

This study was designed to evaluate the PSAA’s prevalence, location, diameter, and visibility in a Saudi population using CBCT imaging, with the goal of improving safety and precision in maxillary sinus surgery. Understanding the artery’s radiographic characteristics in specific populations can contribute to the development of population-based anatomical reference data and improve preoperative risk assessment protocols.

## 2. Materials and Methods

### 2.1. Study Design and Sample Selection

This retrospective cross-sectional study was carried out at Taibah University Dental Hospital (TUDH), Al-Madinah Al-Munawarah, Saudi Arabia, utilizing archived cone-beam computed tomography (CBCT) scans to assess the anatomical characteristics of the posterior superior alveolar artery (PSAA). A total of 117 CBCT scans were selected, comprising 234 maxillary sinuses (right and left), from patients who had undergone dental treatment at TUDH. No formal sample-size calculation was performed because all eligible CBCT scans obtained during the study period were included, ensuring comprehensive data coverage rather than a pre-defined target number.

The inclusion criteria required Saudi patients with high-resolution CBCT images that show complete visualization of the maxillary sinus anatomy. Exclusion criteria included a history of maxillary sinus surgery or trauma involving the sinus wall, presence of sinus pathology that could alter normal morphology (e.g., cysts, neoplasms, or severe mucosal thickening, mild mucosal thickening (<5 mm) was accepted), as it does not obscure PSAA visualization, and CBCT images with insufficient diagnostic quality due to artifacts, scatter, incomplete sinus visualization, or suboptimal exposure settings.

### 2.2. CBCT Acquisition Parameters

All images were obtained using a KaVo 3D eXam CBCT system (KaVo Dental GmbH, Biberach, Germany) under standardized acquisition protocols with a voxel resolution of 0.3 × 0.3 mm^2^. Other radiographic parameters, including kilovoltage peak (kVp), milliampere (mA), and exposure time, were adjusted according to patient size to ensure optimal image quality with minimal radiation exposure.

### 2.3. Radiographic Assessment and Measurements

Each maxillary quadrant was evaluated separately to ensure accurate anatomical delineation. In cases with retained dentition, the posterior superior alveolar artery (PSAA) position was evaluated with respect to the first and second premolars and molars, using the long axis of each tooth as a reference. For edentulous regions, the long axis of the alveolar crest was built to provide a consistent anatomical reference framework for measuring, and the approximate position of the missing teeth was deduced from the alignment of neighboring or opposing teeth.

The primary outcomes of this study included multiple linear and categorical measurements obtained from coronal CBCT sections. These outcomes included measuring the fol-lowing in the coronal section: the distance from the artery to the (A) ridge crest, (B) floor of the maxillary sinus, (C) medial wall of the sinus, (D) height of the alveolar ridge, and (E) the diameter of the artery; finally, the presence or absence of the septa in the sinus (Figure 1). Also, the PSAA’s anatomical location can be categorized as Type A (on the lateral sinus wall’s outer cortex), Type B (intraosseous), or Type C (beneath the Schneiderian membrane) (Figure 2). All outcomes were recorded separately for premolar and molar regions based on root location or estimated axis in edentulous sites, as shown in Figure 1 and Figure 2.

These variables were also analyzed to investigate potential anatomical variations. The recorded parameters included age, gender, edentulous ridge condition, presence of maxillary sinus pathology, canal detectability (unilateral or bilateral), and the sinus side (left or right).

### 2.4. Examiner Calibration and Reliability

Two calibrated examiners independently performed all radiography measurements using a digital caliper prior to the study’s start (Mardinger et al., 2007) [12]. They also underwent a strict and standardized training and calibration process to ensure consistency in the measurement and interpretation of PSAA at CBCT scans. Both theoretical and practical training sessions were part of the process. Cohen’s kappa coefficient was used to assess inter-examiner reliability, and inclusion in the study required a minimum κ = 0.80. After two weeks, 20% of the cases for each examiner had their intra-examiner variability checked again. The intraclass correlation coefficient (ICC) was used to measure the degree of agreement, and a correlation coefficient of 94% indicated a high degree of agreement between observers.

### 2.5. Statistical Analysis

Data analysis was performed using IBM SPSS Statistics version 26.0 (IBM Corp., Armonk, NY, USA). Descriptive statistics, including mean, standard deviation, and frequency distributions, were calculated for all variables. The Shapiro–Wilk test was used to assess normality. The Kruskal–Wallis test was applied to compare age across PSAA diameter categories. The Mann–Whitney U test was used for comparing continuous variables between the right and left sides. Spearman’s rank correlation coefficient (r_s_) was used to evaluate associations between PSAA measurements (distances and diameters) and demographic factors such as age and smoking status. The Chi-square test with Monte Carlo simulation was used to analyze associations between categorical variables, including smoking and artery diameter. A *p*-value < 0.05 was considered statistically significant.

### 2.6. Ethical Approval

The Research Ethics Committee at Taibah University granted approval for this study (Approval No. TUCDREC/011024/AAlqhtani, approved on 11 October 2024). All CBCT images were anonymized, and patient data were handled in compliance with institutional and national ethical guidelines. Informed consent was waived by the Research Ethics Committee at Taibah University, College of Dentistry (TUCD-REC).

## 3. Results

Excellent reliability was confirmed for all linear measurements (intra-observer ICC = 0.94; inter-observer κ = 0.88). A total of 117 CBCT scans were selected for the evaluation of PSAA. Radiographic mapping of the PSAA was performed using the coronal view of CBCT scans. The artery’s location was related to the alveolar crest, sinus floor, and medial and lateral sinus walls were measured in millimeters. Classifications based on the diameters of PSAA and its mediolateral positions to the maxillary sinus were established.

The demographic data of the included subjects are summarized in Table 1. Participants were grouped as ≤30 years (18.8%), 31–50 years (42.7%), 51–70 years (32.5%), and >70 years (6.0%). Among subjects > 60 years (*n* = 7), 28.6% showed small (<0.8 mm) or undetectable PSAA, whereas all younger groups exhibited arteries ≥ 0.9 mm. Age-group analysis revealed that smaller or absent arteries occurred predominantly among individuals > 60 years. Of the total sample, 81.2% were male, and 18.8% were female. Most subjects (73%) were smokers, 99.1% were dentate, and 33.3% showed mild maxillary sinus pathology. Mild mucosal thickening was observed in 33.3% of sinuses; these cases were included because the vascular course remained clearly visible.

The PSAA was detected in 98.3% of the CBCT scans, and the maxillary sinus showed the presence of intra-bony septa in 37.6% of the subjects on the right side and 29.9% on the left side. The mean alveolar crest height in premolar regions for the right and left sides was 15.64 ± 5.84 and 15.11 ± 5.61, respectively, while for molar regions, it was 10.93 ± 4.34 for the right side and 10.62 ± 3.86 for the left side, with no significant difference between the right and left sides. However, significant differences were observed between premolar and molar regions on both sides, as the *p*-value was less than 0.05, as shown in Table 2 and Table 3.

The PSAA diameter was categorized into three classes according to Mardinger et al. (2007): Class A (≤1.0 mm), Class B (1.0–2.0 mm), and Class C (>2.0 mm) [12]. The distribution of the classes in this study was as follows: A was represented by 58.1%, B was 35.9%, and C was 4.3% for the right side, whereas the left side showed 49.6% for class A, 47% for class B, and 2.6% for class C. Although there was no significant difference between sides, most subjects fell into Class A on both the right and left sides, as shown in Table 2.

The mediolateral position of the PSAA was categorized into three categories: A, the PSAA was found on the cortex of the lateral sinus wall; B, the PSAA was found intraosseous; and C, the PSAA was found below the Schneiderian membrane. For the premolar region at the right side, A was represented by 9.4%, B by 46.2%, and C by 41.9%, while the left side showed 5.1%, 53.8% and 38.5% for A, B, and C, respectively, with no significant difference between the right and left sides.

The molar region showed 6%, 35.9%, and 56.4% for A, B, and C, respectively, on the right side, and for the left sides, the values were 6%, 40.2% and 53%. While no significant difference was found between the right and left sides, the comparison of the position showed significant differences between the molar and the premolar regions, as shown in Table 3.

The distance between the PSAA and the alveolar crest was measured at the premolar and molar regions for the right and left sides. The results showed that there was no significant difference between the sides, but significant differences were observed between the premolar and molar regions. The mean distances from the PSAA to the crest in the premolar region were 21.06 ± 6.02 mm on the right side and 20.74 ± 6.55 mm on the left, while for the molar region, the means were 16.17 ± 4.0 and 15.64 ± 4.35 for the right and left sides, respectively, as shown in Table 4 and Figure 3.

The position of the PSAA within the maxillary sinus was assessed by measuring the distance to the floor and medial wall of the maxillary sinus. The results showed that the mean distances from the PSAA to the sinus floor in the premolar and molar regions on the right side were 5.91 ± 4.16 mm and 6.18 ± 3.47 mm, respectively, while for the left side they were 5.86 ± 3.97 and 6.31 ± 3.45. The distances from the PSAA to the medial sinus wall were 8.25 ± 3.81 mm (premolar) and 13.02 ± 3.53 mm (molar) on the right side and 8.54 ± 3.76 mm (premolar) and 13.06 ± 3.55 mm (molar) on the left side, as shown in Table 4 and Figure 4.

### 3.1. Correlations Between PSAA and Different Diameters

#### 3.1.1. Correlations Between Age and PSAA

Diameter of PSAA

The Kruskal–Wallis test was used to compare age across PSAA diameter categories. For the right side, larger artery diameters (>2 mm) were more common in younger individuals, though statistical significance was not reached (*p* = 0.504 and H (Kruskal—Wallis test) = 2.343). Smaller or absent arteries were observed in older individuals, but the sample sizes for these groups were small, with *p* = 0.642 and H = 1.678. For the left side, as at the right side, PSAA with diameters more than 2 at the left side is more common in younger individuals, even though it is not statistically significant (H = 1.678, *p* = 0.642), as shown in Table 5.

Vertical Distance from PSAA to Alveolar Crest

At the premolar region, a weak negative, non-statistically significant correlation was observed on both sides (right: rs = −0.143, left: rs = −0.117). While for the molar region, a very weak negative correlation was observed with no significant association on either side (right: rs = −0.070, *p* = 0.454; left: rs = −0.025, *p* = 0.787), as shown in Table 6.

Vertical distance from PSAA to Floor of Sinus

At the premolar region, a negligible correlation on both sides was found with no significant association (right: rs = −0.018, *p* = 0.851; left: rs = −0.042, *p* = 0.661). At the molar region, very weak correlations were found with no significant association (right: rs = −0.041, *p* = 0.667; left: rs = 0.066, *p* = 0.480).

Vertical distance from PSAA to Sinus Medial Wall

At the premolar region, a very weak positive correlation on the left side was found (rs = 0.088, *p* = 0.351), and a negligible correlation was found on the right side (rs = 0.040, *p* = 0.671) with no significant association. In the molar region, a weak negative correlation between age and distance to the medial wall was observed on the right side (rs = −0.233, *p* = 0.012), which was statistically significant. A similar but non-significant trend was noted on the left side.

#### 3.1.2. Correlation Between Smoking and Diameter of PSAA

A statistically significant association was observed between smoking status and PSAA diameter on both sides, based on a Monte Carlo simulation of the Chi-square test. On the right side, non-smokers had higher percentages of larger artery diameters (42.9% for 1–2 mm and 14.3% for >2 mm) compared to smokers (13.6% and 4.5%, respectively), as shown in Figure 5.

#### 3.1.3. Relationship Between the Presence of Septa in the Maxillary Sinus and the Vertical Distances from the Posterior Superior Alveolar Artery (PSAA) to Both the Sinus Floor and the Medial Wall, on Both the Right and Left Sides

The presence of septa influences the vertical positioning of the PSAA within the maxillary sinus. The distances from the PSAA to both the sinus floor and medial wall were greater in sinuses with septa on both sides. This anatomical variation is of considerable importance for surgical planning, particularly in procedures such as sinus floor elevation, as it may affect the risk of vascular complications, as shown in Figure 6.

Finally, the majority of PSAA diameters fell into Class A (≤1 mm). The location of the PSAA under the Schneiderian membrane (class C) was prominent in the molar region, while in the molar region, it was class B, intraosseous location. The presence of septa affects the location of PSAA. The data showed that as the PSAA courses from the molar to the premolar region, it approaches the medial wall and sinus floor while moving farther from the alveolar crest. This reflects the normal anatomical path of the artery. In the molar region, the artery shifted closer to the medial sinus wall with increasing age, particularly on the right side. In the premolar region, a similar age-related shift was noted toward the alveolar crest.

## 4. Discussion

The purpose of this retrospective CBCT-based study was to characterize the anatomical dimensions and spatial orientation of the posterior superior alveolar artery (PSAA) in relation to maxillary sinus landmarks, particularly in the premolar and molar regions. The PSAA was identified in 98.3% of cases, a finding that aligns with the high detection rates previously reported in CBCT-based studies ranging from 74% to over 90%, affirming the superior sensitivity of CBCT in delineating fine vascular structures [14,15].

Most of the observed arteries exhibited a diameter ≤ 1 mm (Class A), with a greater frequency on the right side. Anatomically, the PSAA was most frequently located intraosseously or subjacent to the Schneiderian membrane. These observations support earlier findings by Ilgüy et al. (2013), which reported that the intraosseous course is the most prevalent anatomical variation [14]. Similarly, Güncü et al. (2011) found that the artery was predominantly intraosseous in 47% of cases and beneath the sinus membrane in 38%, which is comparable to the present data [15]. Recent in vivo CBCT-based research by Bernardi et al. (2024) further confirmed these distribution patterns, reporting a 100% detection rate of the intraosseous PSAA with a mean diameter of 1.07 mm in living patients [16]. Their findings reinforce the diagnostic value of high-resolution CBCT in delineating fine vascular structures prior to sinus augmentation.

When comparing the premolar and molar regions, there were statistically significant differences observed. In the molar area, the artery tended to lie closer to both the sinus floor and the medial wall, with a higher likelihood of being submembranous (Class C). This pattern reflects the anatomical course of the PSAA, which traverses obliquely from posterior to anterior [12]. The mean vertical distance between the PSAA and the sinus floor observed in this study (≈8.9 mm) is comparable to previous CBCT studies, such as Danesh-Sani et al. (2017; 8.9 mm) and Radmand et al. (2023; 8.7 mm), confirming similar anatomic trends across populations [17,18]. Clinically, this anatomical route has implications for lateral window sinus augmentation procedures, as inaccurate localization of the artery may result in bleeding complications [4].

Moreover, the presence of sinus septa was associated with greater vertical distances between the PSAA and both the sinus floor and medial wall. While this anatomical configuration may place the artery slightly farther from the surgical field, septa alter the sinus morphology and create irregular bony contours that complicate flap elevation, window design, and instrument access, thereby increasing the overall technical difficulty of sinus-augmentation procedures [19].

No statistically significant differences were identified between the right and left sides regarding PSAA positioning or dimensional parameters, suggesting a relatively symmetrical bilateral anatomical distribution. However, region-specific anatomical differences were observed, underscoring the importance of individualized preoperative planning [13].

This study represents one of the few CBCT-based anatomical evaluations of the PSAA conducted in a Saudi Arabian population. The region-specific and population-based focus enhances the understanding of anatomical variability in Saudi patients—an area underrepresented in the literature. Furthermore, this study evaluates the association between PSAA anatomy and other factors such as age, smoking status, and the presence of sinus septa, providing new insights into potentially modifiable clinical risk variables.

Consistent with the age-group analysis, smaller or absent arteries were predominantly observed in smokers and in participants older than 60 years, suggesting age-related reductions in vascular caliber and medial displacement of the PSAA. These findings align with previous CBCT investigations reporting diminished PSAA diameter in elderly and smoking cohorts, likely attributable to nicotine-induced vasoconstriction and endothelial dysfunction within the maxillary microvasculature [17]. Collectively, these results underscore the importance of individualized preoperative vascular assessment, particularly in older or smoking patients.

### 4.1. Clinical Implications

The findings of this study highlight the importance of preoperative CBCT imaging for surgical planning in the posterior maxilla. To minimize the risk of anatomical complications, individualized assessment and surgical planning are essential for each patient. Given the significant variability in PSAA diameter and location, assuming a ‘standard’ anatomical course is unsafe. Preoperative evaluation helps avoid complications such as excessive bleeding during implant placement and sinus elevation procedures.

The present findings suggest and support the following clinical recommendations.

CBCT should be routinely used to assess PSAA location, diameter, and course prior to performing surgical procedures in the posterior maxilla to avoid vascular complications.Intraosseous arteries are more common in premolar regions, while submucosal arteries are more frequent in molar regions. Surgical plans should reflect these regional differences.The presence of sinus septa alters the PSAA’s position and increases surgical complexity, particularly during lateral window sinus elevation.Age-related anatomical changes, such as medial displacement of the PSAA, should be considered during treatment planning, especially in older patients.Smoking is associated with smaller PSAA diameters, highlighting the need for thorough imaging in smokers to anticipate vascular challenges.

These findings show how important it is to personalize each patient’s planning and radiographic assessment to avoid complicating the procedures associated with PSAA.

### 4.2. Strengths and Limitations

A major strength of this study is its relatively large sample size and the use of standardized CBCT protocols. These were performed by well-calibrated clinical examiners with high inter- and intra-examiner reliability, ensuring accurate and reproducible measurements. However, the retrospective nature of the CBCT data introduces certain limitations. For example, selection bias may exist, as the included patients may not fully represent the general population. In addition, the very small proportion of edentulous patients (0.9%) limits the generalizability of the results to completely edentulous implant candidates. Furthermore, the sample showed a marked gender imbalance (81.2% male and 18.8% female), which may influence anatomical variability and limit extrapolation to more balanced populations. Additionally, the study lacked real-time or dynamic vascular data on PSAA blood flow. There was also no intraoperative validation to correlate CBCT findings with actual surgical observations. Nevertheless, the study provides strong anatomical data on the PSAA. However, Future studies should incorporate multi-center prospective designs with intraoperative validation to confirm these anatomical findings. The integration of artificial-intelligence-based segmentation and digital navigation systems may further improve the accuracy of vascular mapping and enhance surgical safety during sinus augmentation.

## 5. Conclusions

The posterior superior alveolar artery (PSAA) in the posterior maxilla showed high prevalence and marked anatomical variability, with region-specific differences between the premolar and molar areas. Its proximity to key sinus landmarks, along with the influence of age, smoking, and sinus septa, underscores the need for individualized preoperative CBCT evaluation to minimize intraoperative complications. This study provides region-specific anatomical data for the Saudi population, supporting safer and more predictable sinus-surgery planning.

## Figures and Tables

**Figure 1 diagnostics-15-02999-f001:**
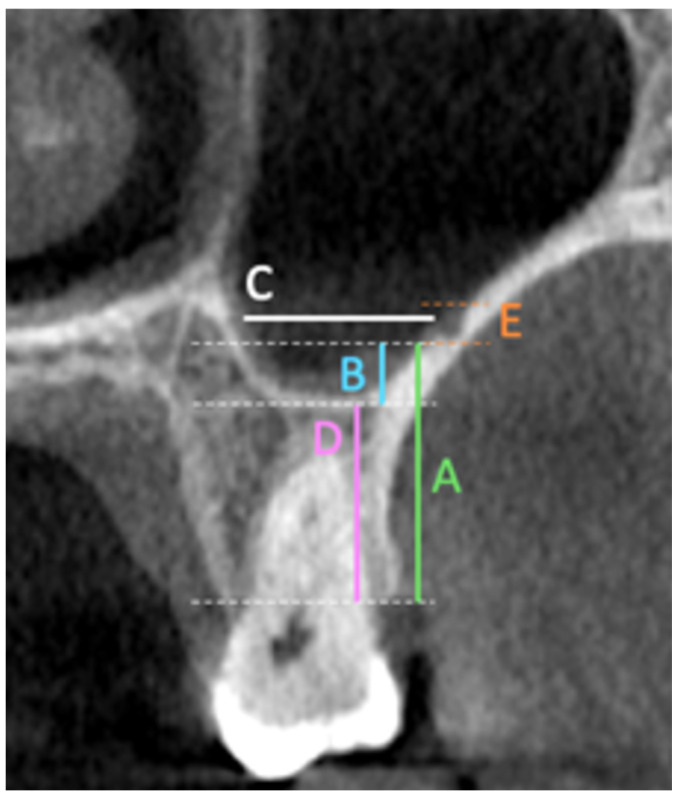
The outcomes included measuring the following in the coronal section: the distance from the artery to the (A) ridge crest; (B) floor of the maxillary sinus; (C) medial wall of the sinus. (D) The height of the alveolar ridge and (E) the diameter of the artery were also measured, as well as the presence or absence of the septa in the sinus. The measurements were taken at two locations corresponding to the premolar and molar regions.

**Figure 2 diagnostics-15-02999-f002:**
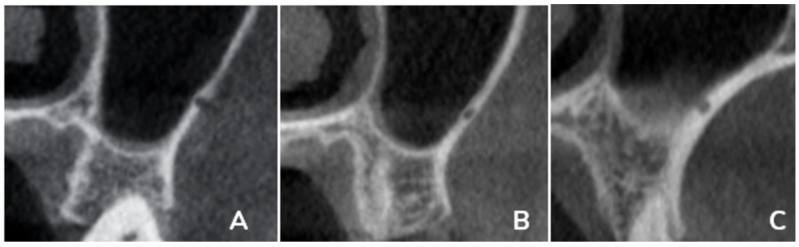
The location of the PSAA is categorized as (**A**) on the outer cortex of the lateral sinus wall, (**B**) intraosseous, or (**C**) below the membrane at locations corresponding to the premolar and molar regions.

**Figure 3 diagnostics-15-02999-f003:**
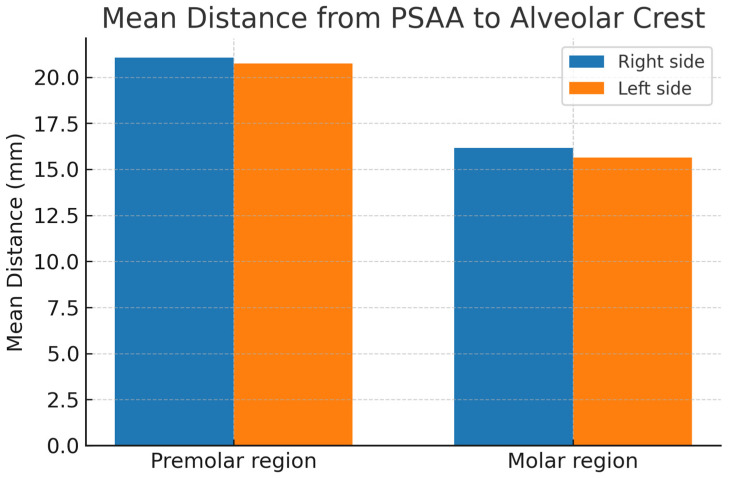
Mean distance from the posterior superior alveolar artery (PSAA) to the alveolar crest in premolar and molar regions on both right and left sides (in millimeters).

**Figure 4 diagnostics-15-02999-f004:**
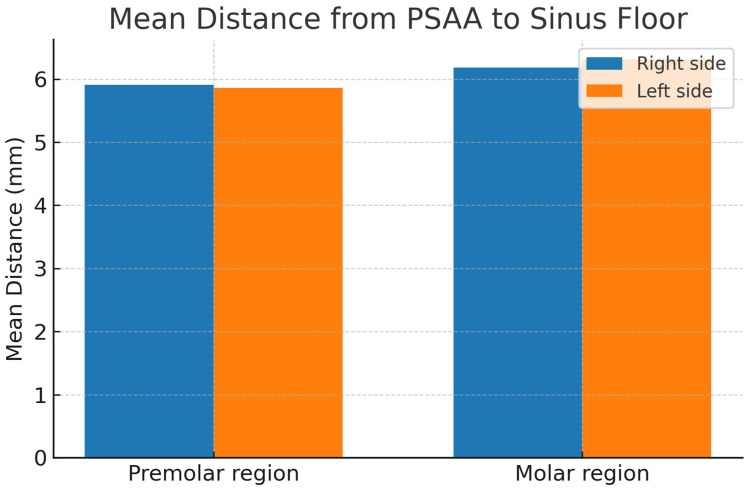
Mean distance from the posterior superior alveolar artery (PSAA) to the sinus floor in premolar and molar regions on both right and left sides (in millimeters).

**Figure 5 diagnostics-15-02999-f005:**
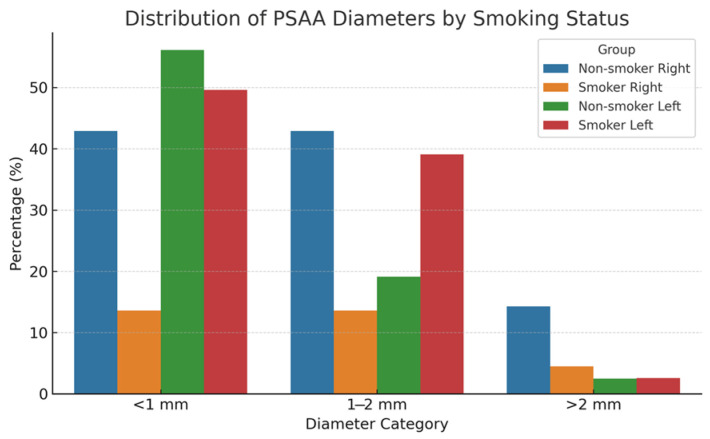
Distribution of PSAA Diameters by Smoking Status.

**Figure 6 diagnostics-15-02999-f006:**
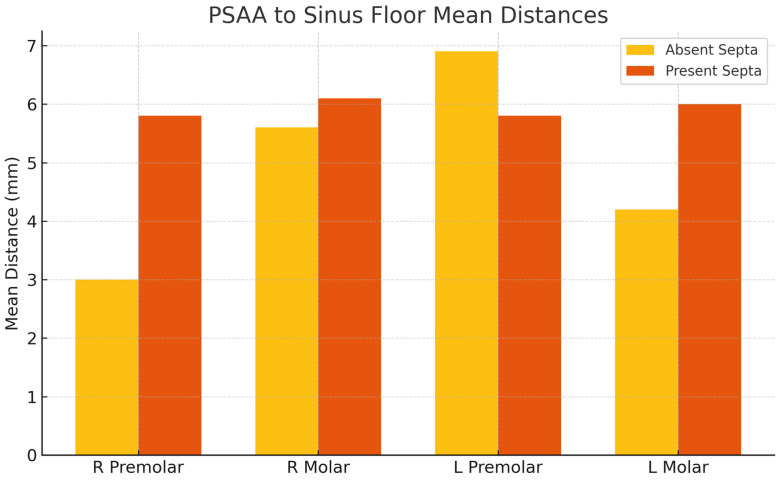
Mean Distance from PSAA to Sinus Floor by Septa Presence.

**Table 1 diagnostics-15-02999-t001:** Demographic Characteristics of the Study Population.

Variable	No. (%)/Value
**Age (years)**	
Min–Max	15.0–94.0
Mean ± SD	41.42 ± 17.47
Median (IQR)	35.0 (31.0–52.0)
**Gender**	
Male	95 (81.2%)
Female	22 (18.8%)
**Smoking Status**	
Non-smoker	44 (37.6%)
Smoker	73 (62.4%)
**Ridge Condition**	
Dentate	116 (99.1%)
Edentulous	1 (0.9%)
**Maxillary Sinus Pathology**	
Present	39 (33.3%)
Absent	78 (66.7%)

**Table 2 diagnostics-15-02999-t002:** Comparison Between Right and Left Side of PSAA.

Variable	Right Side	Left Side	Statistical Test (*p*-Value)
**Presence of PSAA**			
Present	115 (98.3%)	115 (98.3%)	McN test (*p* = 1.000)
Absent	2 (1.7%)	2 (1.7%)	
**Diameter of PSAA (mm)**			
Not detected	2 (1.7%)	1 (0.8%)	MH = 70.000 (*p* = 0.174)
<1 mm	68 (58.1%)	58 (49.6%)	
1–2 mm	42 (35.9%)	55 (47.0%)	
>2 mm	5 (4.3%)	3 (2.6%)	
**Presence of Septa**			
Present	44 (37.6%)	35 (29.91%)	McN test (*p* = 0.064)
Absent	73 (62.4%)	82 (70.09%)	

McN: McNemar test, MH: Marginal Homogeneity Test, *p*: *p* value for comparing between right side and left side.

**Table 3 diagnostics-15-02999-t003:** Comparison of Artery Location and Alveolar Ridge Height Between Right and Left Sides.

Variable	Right Side	Left Side	Statistical Test (*p*-Value)
**Location of Artery—Premolar Region**			
Not present	3 (2.566%)	3 (2.6%)	MH = 108.500 (*p* = 0.908)
Type A (Cortical)	11 (9.401%)	6 (5.1%)	
Type B (Intraosseous)	54 (46.153%)	63 (53.8%)	
Type C (Submucosal)	49 (41.88%)	45 (38.5%)	
**Location of Artery—Molar Region**			
Not present	2 (1.7%)	1 (0.855%)	MH = 120.000 (*p* = 0.837)
Type A (Cortical)	7 (6.0%)	7 (5.983%)	
Type B (Intraosseous)	42 (35.9%)	47 (40.170%)	
Type C (Submucosal)	66 (56.4%)	62 (52.992%)	
MH*p*_0_	0.025 *	0.018 *	
**Alveolar Ridge Height—Premolar Region (** * **n** * ** = 114)**			
Min–Max (mm)	5.0–34.8	3.8–30.0	t = 1.719 (*p* = 0.088)
Mean ± SD	15.64 ± 5.84	15.11 ± 5.61	
Median (IQR)	15 (11.90–18.70)	13.8 (11.6–19.1)	
**Alveolar Ridge Height—Molar Region (** * **n** * ** = 115/116)**			
Min–Max (mm)	2.80–26.60	3.30–24.0	t = 1.214 (*p* = 0.227)
Mean ± SD	10.93 ± 4.34	10.62 ± 3.86	
Median (IQR)	11.1 (8.20–13.15)	10.8 (7.90–12.80)	
*t* * p * _0_	<0.001 *	<0.001 *	

IQR: Interquartile range; SD: Standard deviation; t: Paired *t*-test; MH: Marginal Homogeneity test; *p*: *p*-value comparing right and left sides; *p*_0_: *p*-value comparing premolar and molar regions; *: Statistically significant at *p* ≤ 0.05.

**Table 4 diagnostics-15-02999-t004:** Comparison of PSAA Spatial Position and Distances Between Right and Left Sides.

Variable	Right Side	Left Side	Statistical Test (*p*-Value)
**From PSAA to Alveolar Crest**			
Premolar Region (*n* = 114)			t = 1.109 (*p* = 0.270)
Min–Max (mm)	2.50–36.0	4.0–36.20	
Mean ± SD	21.06 ± 6.02	20.74 ± 6.55	
Median (IQR)	20.0 (16.30–25.30)	20.1 (16.40–25.70)	
Molar Region (*n* = 115/116)			t = 1.555 (*p* = 0.123)
Min–Max (mm)	7.30–30.40	4.90–33.40	
Mean ± SD	16.17 ± 4.00	15.64 ± 4.35	
Median (IQR)	15.7 (13.90–18.30)	15.45 (13.45–17.80)	
t*p*_0_	<0.001 *	<0.001 *	
**From PSAA to Sinus Floor**			
Premolar Region (*n* = 114/113)			Z = 0.419 (*p* = 0.675)
Min–Max (mm)	−3.80–21.30	0.0–20.80	
Mean ± SD	5.91 ± 4.16	5.86 ± 3.97	
Median (IQR)	5.0 (3.20–8.10)	5.20 (3.20–7.40)	
Molar Region (*n* = 115/116)			Z = 0.939 (*p* = 0.348)
Min–Max (mm)	0.0–19.70	0.0–23.70	
Mean ± SD	6.18 ± 3.47	6.31 ± 3.45	
Median (IQR)	5.50 (3.75–8.35)	5.95 (4.05–8.25)	
Z*p*_0_	0.431	0.094	
**From PSAA to Medial Wall**			
Premolar Region (*n* = 114)			t = 1.112 (*p* = 0.268)
Min–Max (mm)	0.0–17.10	0.90–19.90	
Mean ± SD	8.25 ± 3.81	8.54 ± 3.76	
Median (IQR)	7.70 (5.20–11.20)	8.25 (5.80–10.70)	
Molar Region (*n* = 115/116)			t = 0.264 (*p* = 0.792)
Min–Max (mm)	4.0–21.10	4.40–23.50	
Mean ± SD	13.02 ± 3.53	13.06 ± 3.55	
Median (IQR)	13.0 (10.95–15.15)	13.25 (10.4–15.65)	
t*p*_0_	<0.001 *	<0.001 *	

IQR: Interquartile range; SD: Standard deviation; t: Paired *t*-test; *p*: *p*-value comparing right and left sides; *p*_0_: *p*-value comparing premolar and molar regions; *: Statistically significant at *p* ≤ 0.05.

**Table 5 diagnostics-15-02999-t005:** Relation Between Age and PSAA Diameter on Right and Left Sides.

Diameter of Artery (mm)	*n*	Age (Min–Max)	Mean ± SD	Median (IQR)	H (*p*-Value)
**Right Side**					
Not present	2	24–94	59 ± 49.50	59 (24–94)	2.343 (*p* = 0.504)
<1 mm	68	16–84	41.49 ± 17.88	34 (30–55)	
1–2 mm	42	19–82	41.93 ± 15.51	40.5 (31–52)	
>2 mm	5	15–38	29.20 ± 8.87	32 (27–34)	
**Left Side**					
Not present	1	29			1.678 (*p* = 0.642)
<1 mm	58	15–94	42.64 ± 19.31	35 (29–57)	
1–2 mm	55	19–84	40.75 ± 15.78	34 (31–49.5)	
>2 mm	3	27–49	34.33 ± 12.70	27 (27–38)	

IQR: Interquartile range; SD: Standard deviation; H: Kruskal–Wallis test statistic; *p*: *p*-value for differences in age among artery diameter groups.

**Table 6 diagnostics-15-02999-t006:** Relation Between Age and PSAA Different parameters on Right and Left Sides.

Parameter	Side	*n*	rs	*p*-Value	Significance
**Vertical distance to alveolar crest (premolar)**	Right	114	−0.143	0.130	
	Left	114	−0.117	0.216	
**Vertical distance to alveolar crest (molar)**	Right	115	−0.070	0.454	
	Left	116	−0.025	0.787	
**Vertical distance to sinus floor (premolar)**	Right	114	−0.018	0.851	
	Left	113	−0.042	0.661	
**Vertical distance to sinus floor (molar)**	Right	115	−0.041	0.667	
	Left	116	0.066	0.480	
**Vertical distance to medial wall (premolar)**	Right	114	0.040	0.671	
	Left	114	0.088	0.351	
**Vertical distance to**	Right	115	−0.233 *	0.012 *	*

rs = Spearman correlation coefficient; * = Statistically significant at *p* ≤ 0.05.

## Data Availability

The original contributions presented in this study are included in the article. Further inquiries can be directed to the corresponding author.

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
