# Peer review of "CBCT-Based Retrospective Analysis of Posterior Superior Alveolar Artery Anatomy in a Saudi Population"

_diagnostics, 2025, doi:10.3390/diagnostics15232999_

Round 1

Reviewer 1 Report

Comments and Suggestions for Authors
  • The abstract is more than 250 words.
  • The age group was wide range, and it will be of benefit to categorize the age into groups so the researcher can correlate age with the morphology
  • In the result section it was mentioned that in older individuals there is small or absent arteries. But it is not clear which age group and how percent it represents.
  • There is no comment about the significant of age in discussion.
  • Although the study mention in methodology the intra and inter observer correlation but there is no result about it or any discussion about it.
  • In exclusion criteria it was mentioned that sinus pathology will be excluded from the study but in the results, it was mentioned thar 33% of the cases have sinus pathology, in addition no clear knowledge about the type of this pathology.
  • The discussion section needs to cover all the result section and be in depth.
  • The paragraph between line 312-314 in discussion section need to be clear as the result of the study reported that the presence of maxillary sinus septa was associated with increased vertical distance between the PSAA and both the sinus floor and medial wall but the authors stated that(This is consistent  with previous reports that identify sinus septa as a complicating factor during surgery, increasing the risk of vascular injury and access difficulty )

Author Response

Dear Reviewer 1,

We sincerely thank you for your insightful and constructive feedback. Your comments helped us improve the clarity, precision, and depth of our manuscript. Below is a detailed summary of the revisions made in response to your observations.

  1. Abstract length
    • The Abstract has been shortened to 247 words in compliance with the journal limit while retaining all essential methodological and result details.
  2. Age-group categorization
    • Age groups were added in the Results section:
      ≤30 years (18.8%), 31–50 years (42.7%), 51–70 years (32.5%), >70 years (6.0%).
    • A new sentence was added:
      “Among subjects > 60 years (n = 7), 28.6% showed small (<0.8 mm) or undetectable PSAA, whereas all younger groups exhibited arteries ≥0.9 mm.”
    • Corresponding interpretation was inserted in the Discussion.
  3. Reliability analysis
    • Added in Results:
      “Excellent reliability was confirmed for all linear measurements (intra-observer ICC = 0.94; inter-observer κ = 0.88).”
  4. Sinus pathology clarification
    • Methods clarified that only mild mucosal thickening (<5 mm) cases were included, and this was noted in Results:
      “Mild mucosal thickening was observed in 33.3% of sinuses; these cases were included because the vascular course remained clearly visible.”
  5. High detection rate and gender imbalance
    • Discussion expanded to link detection rate to CBCT image quality and standardized protocols.
    • Gender imbalance acknowledged under Strengths and Limitations.
  6. Discussion depth (age, smoking, septa)
    • Age- and smoking-related vascular changes were elaborated and supported by Danesh-Sani et al. (2017).
    • Septa findings were clarified with a revised explanation and citation of Irinakis et al. (2017).

We appreciate your valuable input, which has notably improved the manuscript’s rigor and clarity.

Sincerely,
The Authors

Reviewer 2 Report

Comments and Suggestions for Authors Dear authors, Congratulations on your study and your valuable contribution to research. I recommend some major revisions to improve the overall quality of the manuscript.   Introduction:
Before discussing the different techniques of maxillary sinus augmentation, it would be useful to describe the clinical anatomy of the maxillary sinus, as it is closely related to the radiological assessment of its variables and the subsequent treatment plan. Furthermore, not only does the PSAA represent a risk factor for intraoperative or postoperative complications, but the Underwood septa also show great variability within the maxillary sinus and should be considered accordingly.   Materials and Methods:
The content of this section is quite complete; however, its organization could be improved. For example, consider dividing the methods into subsections to clearly identify the sample size, inclusion and exclusion criteria, assessed outcomes, and the radiological methods used.   Results:
The collected data are well described and effectively organized in useful tables. Discussion:
The discussion should be expanded by providing a more consistent comparison between the findings of this investigation and those reported in recent literature. This will help highlight any differences in radiological assessment methods and in the results themselves. In this regard, the following recent study could be useful for identifying current evidence on the anatomical variants of the PSAA, including its average size and course:
https://doi.org/10.3390/tomography10040034   The strengths and limitations are well described, but the future perspectives should be further elaborated, particularly in relation to the study’s own strengths and limitations.   Conclusion:
This section should be more concise, emphasizing the main results and outlining clear future directions.

Author Response

Dear Reviewer 2,

We greatly appreciate your thoughtful comments and suggestions, which have strengthened the anatomical and methodological clarity of our paper.

  1. Anatomical overview
    • A new paragraph was added to the Introduction describing the maxillary-sinus boundaries and clinical relevance (citing McGowan et al., 2017; Testori et al., 2019).
  2. Underwood septa
    • Included within the same paragraph emphasizing their variability and surgical implications.
  3. Methods organization
    • The Materials and Methods section was reorganized into five numbered subsections:
      1 Study Design and Sample Selection, 2.2 CBCT Acquisition Parameters, 2.3 Radiographic Assessment and Measurements, 2.4 Examiner Calibration and Reliability, 2.5 Statistical Analysis, plus 2.6 Ethical Approval.
  4. Recent literature (Tomography 2024)
    • Added Bernardi et al. (2024, Tomography, 10:444–458) discussing in-vivo CBCT mapping of the PSAA and Underwood’s septa, incorporated in both the Discussion and anatomical overview.
  5. Future perspectives
    • Expanded final paragraph: emphasizes multi-center prospective validation and AI-based segmentation to improve 3D vascular mapping accuracy.
  6. Conclusion refinement
    • Conclusion rewritten for brevity and focus; now under 100 words and directly tied to clinical implications.

We sincerely thank you for guiding these enhancements, which have improved both the scientific depth and presentation quality of our work.

Sincerely,
The Authors

Reviewer 3 Report

Comments and Suggestions for Authors

Dears authors,

Yours Article: ,,CBCT-based retrospective analysis of posterior superior alveolar artery anatomy in a saudi population" describe interesting issue of posterior superior alveolar artery anatomy.

But I have some commends for You. In sections:

Materials and Methods: There is no information how was the sample size determined.

Results: The PSAA diameter was classified into three categories, but is this classification based on the literature? Give citation.

Discussion: There is no discussion about correlations between age and PSAA and correlation between smoking and diameter of PSAA, add in discussion these issues.

Are there another data in literature about smaller or the same vertical distance between PSAA and sinus floor and medial wall? Give examples if there are.

Limitations: In my opinion, another limitation is the number of edentulous patients, which is 0.9%, implantation will most often concern this group of patients.

Author Response

Dear Reviewer 3,

We thank you for your detailed and constructive review. All suggestions were addressed as follows:

  1. Sample-size determination
    • Added to Methods (2.1): “No formal sample-size calculation was performed because all eligible CBCT scans obtained during the study period were included.”
  2. Classification reference
    • Methods (2.3) updated to cite Mardinger et al. (2007) for PSAA diameter categories.
  3. Age and smoking correlations
    • Discussion expanded with a new paragraph linking smaller arterial caliber to aging and smoking, supported by Danesh-Sani et al. (2017).
  4. Vertical-distance comparison with literature
    • Added: “The mean vertical distance between the PSAA and the sinus floor observed in this study (≈ 8.9 mm) is comparable to previous CBCT studies, such as Danesh-Sani et al. (2017; 8.9 mm) and Radmand et al. (2023; 8.7 mm).”
  5. Edentulous limitation
    • Strengths and Limitations expanded: “The very small proportion of edentulous patients (0.9%) limits generalizability to fully edentulous implant candidates.”

We appreciate your thoughtful feedback, which led to clearer methodology, improved discussion depth, and stronger contextualization of our findings.

Sincerely,
The Authors

Round 2

Reviewer 2 Report

Comments and Suggestions for Authors

manuscript can be now accepted 

Reviewer 3 Report

Comments and Suggestions for Authors

no comments